# CORRELATION ANALYSIS OF EVALUATION METRICS FOR MACHINE TRANSLATION

## ABSTRACT

Machine translation evaluation methods can be roughly divided into three categories: manual evaluation, classical morphological evaluation and semantic evaluation based on pre-trained model. The automatic evaluation metrics of the latter two categories are numerous, from which we select commonly used seven morphological evaluation metrics and four semantic evaluation metrics for correlation analysis between each two of them. The experimental results of the correlation coefficients of Pearson, Kendall and Spearman on 40 machine translation models of bidirectional 20 foreign languages and Chinese show that: (1) There is an extremely strong correlation among morphological evaluation metrics, indicating that the statistical results of various morphological calculation methods tend to be the same on big data. (2) There is a strong correlation between semantic evaluation metrics, indicating that although there are semantic spatial differences among various pre-trained models, the statistical results on big data also tend to be consistent. The above-mentioned ubiquitous correlations largely stem from the equivalence of human cognition and the economy of knowledge representation. (3) There is also a strong correlation between morphological and semantic evaluation metrics, which shows that the deep "semantics" of various commercial hypes at present is just another high-level "morphology". Because the Turing computing system can use symbols and operations to directly represent and accurately process morphologies, but can only simulately represent and approximately process semantics using symbols and operations. (4) For each correlation coefficient between any two evaluation metrics, there is a significant difference between different languages, which indicates that morphology and semantics are inherent attributes of languages, and more optimized evaluation metrics of machine translation should be personalized according to the language.

## 1 INTRODUCTION

Machine Translation (MT) is an algorithmic computing process that uses a target natural language form to paraphrase the semantics of a source natural language. MT research has puzzled human beings for a long time. It was not until the emergence of deep learning and language big data that the laboratory translation quality of rule-based MT and statistical MT was changed, and almost perfect and usable translation was obtained. Because of this, the research of MT evaluation to judge the quality of translation has been accompanied by MT research and has experienced a long history, resulting in many MT evaluation methods.

Up to now, MT evaluation methods can be roughly divided into three categories: manual evaluation, classical morphological evaluation and semantic evaluation based on pre-trained model.

The input of manual evaluation is usually a sentence pair of *<SSen, TSen.MT>*. Here, the *SSen* denotes a source language sentence, and the *TSen.MT* denotes a machine-translated target language sentence. Human experts give a binary judgment of *GOOD* or *BAD*, or a score belonging to the interval of [0, 100] on the semantic fit of the two sentences according to their own knowledge of language and culture (Liu, 2022).

Manual evaluation is not only time-consuming and laborious but also has personality deviations among different experts. Therefore, a morphological evaluation method that hardly requires human participation has been proposed. The input of this classical method is a triple of *<SSen, TSen.MT, TSen.M>*. Here, the newly added *TSen.M* denotes a manual-translated reference sentence of the target language. The morphological evaluation method outputs a morphological similarity score belonging to the [0, 1] interval only through the morphological calculation between sentences in the triple, and estimates the semantic fit degree of machine-translated sentence accordingly.

With the successful application of deep learning and language big data, a semantic evaluation method based on pre-trained models has been proposed. The input of this method adds a pre-trained model on

the basis of the input triple of morphological evaluation. Through the pre-trained model, the machine-translated sentence and the manual-translated reference sentence are embedded into two vectors, and then the similarity between the two vectors is calculated, and a semantic similarity score belonging to the [0, 1] interval is output, according to which the semantic fit degree of the machine-translated sentence is directly judged.

The automatic evaluation metrics of the latter two categories are numerous (Lee, 2023), and we hope to re-examine the correlation between these metrics through quantitative analysis, and further elaborate the relationship between morphology and semantics.

## 2 AUTOMATIC EVALUATION METRICS

In this paper, we have chosen commonly used seven morphological evaluation metrics and four semantic evaluation metrics.

### 2.1 MORPHOLOGICAL EVALUATION METRICS

The first morphological evaluation metric we choose is the famous BLEU, which has almost become an internationally recognized standard for MT research papers and application systems. In this paper, the BLEU refers to the classic BLEU4 score. We then choose two morphological evaluation metrics, TER and CHRF, which represent the translation edit rate (Post, 2018) and the character-level chrF2 score (Popović, 2015), respectively. Due to space limitations, we will not repeat the above three well-known morphological evaluation metrics for MT.

Because any morphological similarity between two strings can be regarded as a morphological evaluation metric for MT. Therefore, we also choose four morphological evaluation metrics (Leven, Jaccard, Dice, Cosine) between the machine-translated sentence y' and the manual-translated reference sentence y.

$$\text{Leven} = 1 - \frac{LevenDis(\text{y}, \text{y}')}{Max(Len(\text{y}), Len(\text{y}'))} \quad (1)$$

The Levenshtein morphological evaluation metric (Leven) is based on the edit distance proposed by Soviet mathematician Vladimir Levenshtein in 1965. The edit distance is the minimum number of edit operations required to convert one string to another, including replacements, insertions, and deletions. The Leven is calculated as shown in formula (1), where $LevenDis$(y, y') represents the token-level Levenshtein edit distance between the manual-translated reference sentence y and the machine-translated sentence y', $Len$(y) and $Len$(y') represent the length of the sentence y and the sentence y' respectively, that is, the number of tokens contained in each. Leven values range from 0 to 1, and the higher the value, the more similar the morphology of the two sentences.

$$\text{Jaccard} = \frac{\#(TokenSet(\text{y}) \cap TokenSet(\text{y}'))}{\#(TokenSet(\text{y}) \cup TokenSet(\text{y}'))} \quad (2)$$

The Jaccard morphological evaluation metric (Jaccard) is based on the proportion of commonality between finite sample sets, that is the ratio of the size of the intersection of two sets to the size of the union of the two sets. When both sets are empty, the Jaccard value is defined as 1. This statistic used for gauging the similarity and diversity of sample sets was developed by American geologist Grove Karl Gilbert in 1884. The Jaccard is calculated as shown in formula (2), where $TokenSet$(y) and $TokenSet$(y') respectively represent the token set obtained after the tokenization process of the manual-translated reference sentence y and the machine-translated sentence y', and #(·) represents the number of elements in the set. Jaccard values range from 0 to 1, and the higher the value, the more similar the morphology of the two sentences.

$$\text{Dice} = \frac{2 * \#(TokenSet(\text{y}) \cap TokenSet(\text{y}'))}{\#(TokenSet(\text{y})) + \#(TokenSet(\text{y}'))} \quad (3)$$

The Dice morphological evaluation metric (Dice), also known as Sørensen-Dice metric, is a measure of set similarity published independently by American ecologist and geneticist Lee Raymond Dice and Danish botanist Thorvald Sørensen in 1945 and 1948, respectively. The Dice is calculated as shown in formula (3), where the definitions of $TokenSet$(y), $TokenSet$(y'), and #(·) are the same as in formula (2). Dice values also range from 0 to 1, the higher the value, the more similar the morphology of the two sentences.

$$\text{Cosine} = \frac{TokenFreqVec(\text{y}) \cdot TokenFreqVec(\text{y}')}{||TokenFreqVec(\text{y})||\,||TokenFreqVec(\text{y}')||} \quad (4)$$

The cosine morphological evaluation metric (Cosine) is the classical cosine value between two token frequency vectors. The Cosine is calculated as shown in formula (4), where **TokenFreqVec**(y) and **TokenFreqVec**(y') represent the token frequency vectors of sentences y and y', respectively. Because the token frequency is always non-negative, the value of Cosine belongs to [0, 1], and the higher the value, the more similar the morphology of the two sentences.

## 2.2 SEMANTIC EVALUATION METRICS

Similar to the morphological evaluation metrics, we can also implement the semantic evaluation metrics by calculating the semantic similarity between two sentences in the same language. Specifically, first, we use the Sentence Transformer[1] to generate two embedding vectors of **Embed**(m, y') and **Embed**(m, y) from the machine-translated sentence y' and the manual-translated reference sentence y supported by the pre-trained model m (Peters, 2018). Secondly, the cosine similarity between the two sentence embedding vectors in the pair <**Embed**(m, y'), **Embed**(m, y)> is calculated in the semantic evaluation metric. Finally, the similarity is normalized to the [0, 1] interval according to the formula (5) as the semantic similarity (SS) score between the two sentences in the sentence pair <x, y>. Any Sentence-BERT series of pre-trained models[2] that support sentence embedding can be used for the SS score calculation. The serial models use siamese and triplet network structures to derive semantically meaningful sentence embeddings. This representation easily supports cosine similarity calculation and greatly reduces computational overhead while maintaining the accuracy of BERT.

$$SS(m, y', y) = \frac{CosSim(\textbf{Embed}(m, y'), \textbf{Embed}(m, y)) + 1}{2} \quad (5)$$

Table 1 provides an overview of four pre-trained models all-distilroberta-v1, all-MiniLM-L6-v2, all-mpnet-base-v2, and all-roberta-large-v1. They are all trained on a large and diverse dataset of over 1 billion training pairs, and they are all all-round model tuned for many use cases, which can be directly used in semantic evaluation metrics. We abbreviate the semantic evaluation metrics of the corresponding models according to the formula (5) as Distil = SS(all-distilroberta-v1,y',y), MiniLM = SS(all-MiniLM-L6-v2,y',y), Mpnet = SS(all-mpnet-base-v2,y',y), and Roberta = SS(all-roberta-large-v1,y',y), respectively.

Table 1: Pre-trained Model Overview

| Pre-trained Model (Base Model) | Max Sequence Length | Dimensions | Model Size (MB) | Speed | Sentence Embedding Performance | Semantic Search Performance | Avg. Performance |
|---|---|---|---|---|---|---|---|
| **all-distilroberta-v1** (distilroberta-base) | **512** | 768 | 290 | 4,000 | 68.73 | 50.94 | 59.84 |
| **all-MiniLM-L6-v2** (nreimers/MiniLM-L6-H384-uncased) | 256 | 384 | **80** | **14,200** | 68.06 | 49.54 | 58.80 |
| **all-mpnet-base-v2** (microsoft/mpnet-base) | 384 | 768 | 420 | 2,800 | 69.57 | **57.02** | **63.30** |
| **all-roberta-large-v1** (roberta-large) | 256 | **1,024** | 1,360 | 800 | **70.23** | 53.05 | 61.64 |

Among the four pre-trained models mentioned above: The all-distilroberta-v1 is a smaller pre-trained general-purpose language representation model. During its pretraining phase, knowledge distillation is leveraged to reduce the size of a BERT model by 40%, while retaining 97% of its language understanding capabilities and being 60% faster (Sanh, 2020). The all-MiniLM-L6-v2 is a deep self-attention distillation model that uses the formula (6) to minimize the KL difference between the self-attention distributions of teachers and students and can effectively compress pre-trained models based on large Transformers (Wang, 2020). The monolingual model of the deep self-attention distillation method outperforms the optimal baseline under different parameter sizes of the student model. The all-mpnet-base-v2 model that can see a full sentence leverages the dependency among predicted tokens through permuted language modeling and takes auxiliary position information as input, which can reduce the position discrepancy, and outperform masked language modeling and permuted language modeling by a large margin (Song, 2020). The all-roberta-large-v1 is an improved BERT model based on a replication study of the BERT hyperparameter choices, which can achieve state-of-the-art results on GLUE, RACE, and SQuAD (Liu, 2019).

$$L_{AT} = \frac{1}{A_h|x|} \sum_{a=1}^{A_h} \sum_{t=1}^{|x|} D_{KL} \left( A_{L,a,t}^T || A_{M,a,t}^S \right) \quad (6)$$

The four pre-trained models in Table 1 have been extensively evaluated for their quality to embedded sentences (Sentence Embedding Performance) and to embedded search queries & paragraphs (Semantic

---

[1] https://www.sbert.net
[2] https://huggingface.co/models

Search Performance). The Sentence Embedding Performance is average performance on encoding sentences over 14 diverse tasks from different domains. The Semantic Search Performance is performance on 6 diverse tasks for semantic search: encoding of questions and paragraphs up to 512 word pieces. While the Avg. Performance is the average of Sentence Embedding Performance and Semantic Search Performance. The higher the value of all three performance metrics, the better the performance. The data in Table 1 show that the all-distilroberta-v1 model has the longest Max Sequence Length (512); the all-mpnet-base-v2 model provides the best quality (63.30) in Avg. Performance; the all-roberta-large-v1 model provides the best quality (70.23) in Sentence Embedding Performance; while the all-MiniLM-L6-v2 model is the smallest (80MB) but has the fastest speed (14,200), 3.5 times the speed of all-distilroberta-v1, 5 times the speed of all-mpnet-base-v2, and more than 17 times the speed of all-roberta-large-v1 and still offers good quality.

# 3 CORRELATION ANALYSIS

Correlation analysis usually refers to the numerical statistics of two or more interrelated variables, so as to measure the closeness of the correlation between variables. Among the many correlation analysis methods, the Pearson correlation coefficient method, the Kendall correlation coefficient method, and the Spearman correlation coefficient method are widely used to measure the correlation between two variables. We use the above three correlation coefficient methods to quantitatively analyze the correlation between any two of the 11 automatic evaluation metrics in the previous section.

## 3.1 FRAMEWORK

Figure 1 shows our proposed correlation analysis framework, which mainly includes two machine translators (XZho Machine Translator and ZhoX Machine Translator) that support bidirectional translation between language X and Chinese, one group of 7 morphological evaluation metrics (Leven, Jaccard, Dice, Cosine, TER, CHRF, and BLEU), one group of 4 semantic evaluation metrics based on pre-trained models (Distil, MiniLM, Mpnet, and Roberta), and three correlation analyzers (Pearson Correlation Analyzer, Kendall Correlation Analyzer, and Spearman Correlation Analyzer).

When a set of sentence pairs arrives, each pair of sentences <XSen, ZhoSen> is taken out in turn, and XSen and ZhoSen are processed in parallel. For the sentence XSen, first, the sentence is sent to the XZho Machine Translator, and the translated sentence is ZhoSen.MT. Then, the calculating units of the 7 morphological evaluation metrics and the 4 semantic evaluation metrics receive ZhoSen.MT and ZhoSen concurrently, and synchronously calculate and output 11 scores belonging to the [0, 1] interval (Leven, Jaccard, Dice, Cosine, TER, CHRF, BLEU, Distil, MiniLM, Mpnet, and Roberta). Finally, the three correlation analyzers receive the above 11 scores respectively, calculate the correlation coefficients between each two kinds of score, and output a visualized heat map. For the sentence ZhoSen, the processing is the same as above, except for the MT direction.

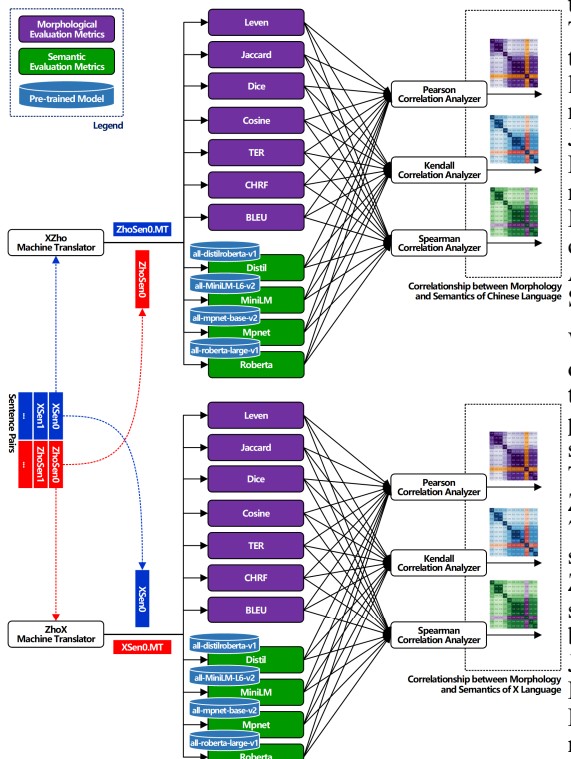

Figure 1: Correlation Analysis Framework

## 3.2 CORRELATION COEFFICIENTS

Pearson correlation coefficient developed by Karl Pearson from a related idea in the 1880s, the ratio between the covariance of two variables $X$ and $Y$ and the product of their standard deviations, is often used to measure the linear correlation between two sets of data $\{x_0, x_1, ..., x_{n-1}\}$ and $\{y_0, y_1, ..., y_{n-1}\}$.

Given $n$ pairs of data $\{(x_0, y_0), (x_1, y_1), ..., (x_{n-1}, y_{n-1})\}$ consisting of the two data sets, the sample Pearson correlation coefficient $r_p$ is defined as formula (7). Where $n$ is the sample size, $x_i, y_i$ are the sample points, and $\bar{x}$, $\bar{y}$ are the sample means. This coefficient is essentially a normalized measurement of the covariance, so its value belongs to [-1, 1]. Where a negative value indicates a negative linear correlation, that is, as the value of one variable increases, the other decreases; A positive value indicates a positive linear correlation, that is, as the value of one variable increases, the other also increases; zero indicates that there is no linear relationship between the two variables.

$$r_p = \frac{\sum_{i=0}^{n-1}(x_i - \bar{x})(y_i - \bar{y})}{\sqrt{\sum_{i=0}^{n-1}(x_i - \bar{x})^2}\sqrt{\sum_{i=0}^{n-1}(y_i - \bar{y})^2}} \qquad (7)$$

The applicable scenario of the Pearson correlation coefficient is a continuous variable with normal distribution. According to the central limit theorem, when the sample size is large enough (exceeds 500), the data can be considered to be approximately normally distributed. But as with covariance itself, the Pearson correlation coefficient can only reflect a linear correlation of variables and ignores many other types of relationships or correlations. That is to say when $r_p = 0$ can only mean that there is no linear relationship between variables, and it is not sure whether there are other correlations. While the other two rank correlation coefficients are not limited by sample size and normal distribution of samples.

Kendall correlation coefficient developed by Maurice Kendall in 1938, a statistic used to measure the ordinal association between two measured quantities, is a rank correlation coefficient. It is a measure of rank correlation: the similarity of the orderings of the data when ranked by each of the quantities. Let $(x_0, y_0), (x_1, y_1), ..., (x_{n-1}, y_{n-1})$ be a set of observations of the joint random variables $X$ and $Y$, such that all the values of $(x_i)$ and $(y_i)$ are unique, the Kendall correlation coefficient $r_k$ is defined as formula (8). Where $n$ is the sample size, $x_i, y_i$ are the sample points, and $sgn(\cdot)$ is the sign function. This definition shows that the value of the Kendall correlation coefficient belongs to [-1, 1]. If the agreement between the two rankings is perfect the coefficient has value 1. If the disagreement between the two rankings is perfect the coefficient has value -1. If $X$ and $Y$ are independent random variables and not constant, then the expectation of the coefficient is zero.

$$r_k = \frac{2}{n(n-1)}\sum_{i<j} sgn(x_i - x_j)sgn(y_i - y_j) \qquad (8)$$

Spearman correlation coefficient is also a rank correlation coefficient, which is named after the English psychologist Charles Spearman. The Spearman correlation between two variables is equal to the Pearson correlation between the rank values of those two variables. For a sample of size $n$, the $n$ raw scores $x_i, y_i$ are converted to ranks $rank(x_i), rank(y_i)$. Then the two new variables $rank(x_i)$ and $rank(y_i)$ are brought into formula (7) to obtain the Spearman correlation coefficient $r_s$ shown in formula (9). Where $rank(\cdot)$ is the ranking function that maps the original score to a positive integer, so the sample mean $\overline{rank(x)} = \overline{rank(y)}$. Based on the $rank(\cdot)$ function, the Spearman correlation coefficient $r_s$ can be reduced to formula (10).

$$r_s = \frac{\sum_{i=0}^{n-1}(rank(x_i) - \overline{rank(x)})(rank(y_i) - \overline{rank(y)})}{\sqrt{\sum_{i=0}^{n-1}(rank(x_i) - \overline{rank(x)})^2}\sqrt{\sum_{i=0}^{n-1}(rank(y_i) - \overline{rank(y)})^2}} \qquad (9)$$

$$r_s = 1 - \frac{6\sum_{i=0}^{n-1}(rank(x_i) - rank(y_i))^2}{n(n^2 - 1)} \qquad (10)$$

Among the above three correlation coefficients, the Pearson correlation coefficient $r_p$ focuses on measuring the linear correlation between the original variables, the Kendall correlation coefficient $r_k$ focuses on measuring the rank correlation between the original variables, and the Spearman correlation coefficient $r_s$ focuses on measuring the linear correlation between the ranks of the original variables. Therefore, the prerequisite for the Pearson correlation coefficient is the highest, and the Kendall correlation coefficient is more suitable for relatively ordered variables, while the Spearman correlation coefficient is not sensitive to outliers and is more suitable for any type of variable. The strength of the correlation is determined by the value of $|r|$. The usual 5 grades include: extremely weak correlation ($|r| \in [0.00, 0.19]$), weak correlation ($|r| \in [0.20, 0.39]$), moderate correlation ($|r| \in [0.40, 0.59]$), strong correlation ($|r| \in [0.60, 0.79]$), extremely strong correlation ($|r| \in [0.80, 1.00]$). There are also 3 grades including: weak correlation ($|r| \in [0.10, 0.30]$), moderate correlation ($|r| \in (0.30, 0.50)$), strong correlation ($|r| \in [0.50, 1.00]$).

# 4 EXPERIMENT

In order to quantitatively analyze the correlation between morphological and semantic evaluation metrics of multiple languages, we first implemented a bidirectional MT experimental system between 20 foreign languages and Chinese, which includes 40 neural MT (NMT) models. Secondly, we prepared 20 datasets of XZho sentence pairs with a capacity of 200,000 pairs respectively. Where X belongs to the above set of 20 foreign languages, and Zho denotes Chinese. Thirdly, with the support of MT from the 20 languages to Chinese in the upper half of Figure 1, we carry out a correlation analysis experiment based on Chinese text and obtain the characteristics of Chinese. Finally, with the support of MT from Chinese to the 20 languages in the lower half of Figure 1, we carry out correlation analysis experiments based on texts in various languages and obtain the characteristics of these languages.

## 4.1 MT EXPERIMENTAL SYSTEM

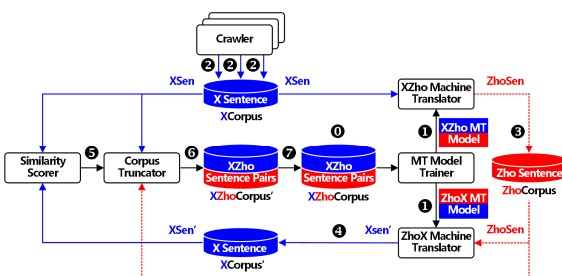

Figure 2: Multiloop Incremental Bootstrapping Framework

Considering the diversity of object languages and the generalization of results, we use the industrial-grade multiloop incremental bootstrapping (MIB) idea to implement multilingual MT experimental system (Liu, 2023). This idea of semi-supervised incremental learning data augmentation idea, which promotes the advantages of supervised learning (Liu, 2017) and unsupervised learning (Liu, 2016), firstly uses appropriate bilingual corpus to train good bidirectional MT models; secondly fully taps the potential of Internet monolingual big data, and uses the trained MT models to translate monolingual sentences twice to incrementally construct a bilingual pseudo-corpus; thirdly the bilingual pseudo-corpus is used to enhance the initial bilingual corpus; Finally, the above process is loop-repeated based on the enhanced bilingual corpus, until the trained MT model meets the optimal performance requirements.

Figure 2 shows the MIB framework, which mainly includes an **MT Model Trainer**, two machine translators (**XZho Machine Translator**, **ZhoX Machine Translator**), several **Crawler**s, a **Similarity Scorer**, and a **Corpus Truncator**. The MIB route is made up of multiple improvement loops. **Step❶**: We need to prepare an XZho (language X to Chinese) corpus of sentence pairs named XZhoCorpus. **Step❶**: The **MT Model Trainer** receives the XZhoCorpus, and trains out two MT models respectively from language X to Chinese and from Chinese to language X. **Step❷**: A group of parallel **Crawler**s continuously crawl language X texts from the Internet, and build a super-large-scale language X sentence corpus (XCorpus). **Step❸**: The **XZho Machine Translator** translates each language X sentence (XSen) in XCorpus into the Chinese sentence (ZhoSen) according to the XZho MT model, and collects them to form a Chinese sentence corpus (ZhoCorpus). **Step❹**: The **ZhoX Machine Translator** translates the Chinese sentence (ZhoSen) in ZhoCorpus back into the language X sentence (XSen') according to the ZhoX MT model, and collects them to form a language X sentence corpus (XCorpus'). **Step❺**: The **Similarity Scorer** calculates the similarity between the source sentence XSen and the result sentence XSen' flowing through the two machine translators. **Step❻**: The **Corpus Truncator** sorts the corresponding sentence pair <XSen, ZhoSen> according to the similarity between XSen and XSen', and truncates the TopN sentence pairs with the highest similarity to form a new XZho corpus of sentence pairs (XZhoCorpus'). **Step❼**: The XZhoCorpus' is merged into the XZhoCorpus. The first closed loop is completed from the Step❶ to the Step❼, and then the second loop is started from the Step❶ again, and so on. The above multiple loops are used together to implement the complete MIB framework.

According to the framework in Figure 2, we use an open-source training module of sequence-to-sequence NMT model[3] to implement the **MT Model Trainer**. The hparams of the NMT model mainly include the number of neurons (*num_units* = 512), the number of encoding and decoding

---

[3] https://github.com/tensorflow/nmt

layers (*num_encoder_layers* = *num_decoder_layers* = 4), the batch size (*batch_size* = 512), and the beam search width (*beam_width* = 10), while others remain the default values. The 20 languages of Arabic (Ara), Czech (Ces), English (Eng), Filipino (Fil), French (Fra), Indonesian (Ind), Italian (Ita), Kazakh (Kaz), Khmer (Khm), Kyrgyz (Kir), Lao, Malay (Msa), Myanmar (Mya), Polish (Pol), Russian (Rus), Slovak (Slk), Spanish (Spa), Thai (Tha), Ukrainian (Ukr), Vietnamese (Vie) are selected and their morphological processing tools are implemented respectively.

Table 2: BLEU Values of NMT Models

| Language Pair | BLEU | Language Pair | BLEU | Language Pair | BLEU | Language Pair | BLEU |
|---|---|---|---|---|---|---|---|
| AraZho | 44.26 | LaoZho | 32.12 | ZhoAra | 35.37 | ZhoLao | 23.08 |
| CesZho | 45.14 | MsaZho | 34.32 | ZhoCes | 34.26 | ZhoMsa | 28.22 |
| EngZho | 48.54 | MyaZho | 32.60 | ZhoEng | 39.23 | ZhoMya | 32.55 |
| FilZho | 45.74 | PolZho | 44.85 | ZhoFil | 30.51 | ZhoPol | 34.39 |
| FraZho | 47.59 | RusZho | 40.92 | ZhoFra | 36.88 | ZhoRus | 34.17 |
| IndZho | 45.90 | SlkZho | 44.79 | ZhoInd | 39.19 | ZhoSlk | 35.28 |
| ItaZho | 41.57 | SpaZho | 47.83 | ZhoIta | 34.65 | ZhoSpa | 37.12 |
| KazZho | 38.26 | ThaZho | 38.95 | ZhoKaz | 28.75 | ZhoTha | 32.79 |
| KhmZho | 37.77 | UkrZho | 44.94 | ZhoKhm | 27.62 | ZhoUkr | 33.37 |
| KirZho | 35.03 | VieZho | 38.51 | ZhoKir | 26.55 | ZhoVie | 32.05 |

We fixed the total number of loops and the increment of sentence pairs (TopN) to 11 and 1,000,000 respectively. The corpus of sentence pairs (XZhoCorpus) for each language and Chinese, that is, the initial training set, contains 5,000,000 sentence pairs, while the final training set will contain 15,000,000 sentence pairs after the execution of 11 loops. We also equip an additional 100,000 sentence-pair development set and 100,000 sentence-pair test set for each language. For each language, the initial training set is the same distribution as the development set and the test set, which are divided from the same corpus by simple random sampling. While the **Crawler** captures from the open domain to form the monolingual sentence corpus (XCorpus), which is independent of the initial training set. To ensure the high availability of the Top1,000,000 pseudo-corpus, monolingual sentences at least 10 times TopN are captured in each loop, and then the Top1,000,000 sentence pairs are truncated based on the Levenshtein similarity score. The BLEU values of bidirectional NMT models[4] between the 20 languages and Chinese are shown in Table 2. Among them, the BLEU values of the English-Chinese and Chinese-English NMT models are the highest, with 48.54 and 39.23, respectively. The BLEU values of the Lao-Chinese and Chinese-Lao NMT models are the lowest, at 32.12 and 23.08, respectively.

## 4.2 EXPERIMENTAL RESULTS IN CHINESE

We run MT and correlation analysis experiments on the 20 datasets of XZho sentence pairs. For each pair of sentences <XSen, ZhoSen> in each XZho dataset, in Chinese space, we calculate the values of the 7 morphological evaluation metrics and the 4 semantic evaluation metrics between the Chinese sentence ZhoSen and the Chinese translation sentence ZhoSen.MT from XSen sentence. And then we perform Pearson correlation analysis on these 11 variables and calculate the Pearson correlation coefficients between any two variables in the {Distil, MiniLM, Mpnet, Roberta, Leven, Jaccard, Dice, Cosine, TER, CHRF, BLEU} set respectively. The 20 Pearson correlation heatmaps shown in Figure 3 (A) were finally drawn.

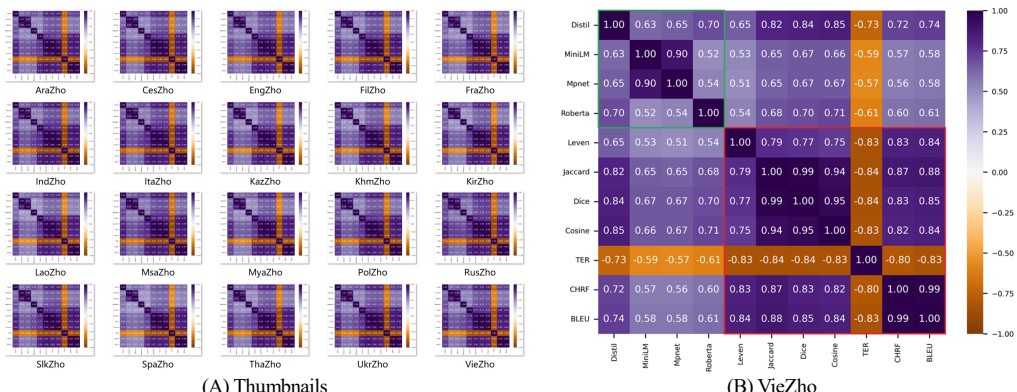

(A) Thumbnails      (B) VieZho

Figure 3: Pearson Correlation Coefficient Heatmap in Chinese

From Figure 3 (A), we can see that the statistical results on the 20 datasets of sentence pairs are basically the same. Without losing generality, we selected the correlation heatmap of VieZho (Vietnamese-Chinese) shown in Figure 3 (B) from the thumbnails for detailed analysis. First, a

---

[4] Anonymous due to review requirements

conspicuous orange cross against a purple grid background, which shows that except for the smaller the TER value, the better the performance of the MT model, the other 10 variables are all the larger the value, the better the performance of the MT model. Secondly, the red 7×7 grids on the lower right of Figure 3 (B) show that there is an extremely strong correlation among morphological evaluation metrics, indicating that the statistical results of various morphological calculation methods tend to be the same on big data. For example, the correlation coefficient of Jaccard and Dice is 0.99, that of CHRF and BLEU is 0.99, that of Dice and Cosine is 0.95, and that of Jaccard and Cosine is 0.94. There is an extremely strong linear correlation in these 4 groups. In particular, the two groups with a correlation coefficient of 0.99 can almost be replaced with each other. Thirdly, the green 4×4 grids on the upper left of Figure 3 (B) show that there is a strong correlation between semantic evaluation metrics, indicating that although there are semantic spatial differences among various pre-trained models, the statistical results on big data also tend to be consistent. For example, the correlation coefficient of MiniLM and Mpnet is 0.90. The above-mentioned ubiquitous correlations largely stem from the equivalence of human cognition and the economy of knowledge representation. Furthermore, it is easy to find that the color grid of the whole figure tends to be dark (purple or orange), and light colors account for a small proportion. That is, the absolute values of all correlation coefficients were greater than 0.5, indicating a strong linear correlation between all 11 variables. The values of correlation coefficients between the morphological evaluation metrics and the semantic evaluation metrics range from 0.51 to 0.85, which shows that the deep "semantics" of various commercial hypes at present is just another high-level "morphology". The Turing computing system can use symbols and operations to directly represent and accurately process morphologies, but can only simulately represent and approximately process semantics using symbols and operations.

Table 3: Average Pearson Correlation Coefficient in Chinese

| | Distil | MiniLM | Mpnet | Roberta | Leven | Jaccard | Dice | Cosine | TER | CHRF | BLEU |
|---|---|---|---|---|---|---|---|---|---|---|---|
| **Distil** | 1.0000 | 0.6140 | 0.6378 | 0.6737 | 0.6334 | 0.7915 | 0.8083 | 0.8221 | -0.7357 | 0.7034 | 0.7234 |
| **MiniLM** | 0.6140 | 1.0000 | 0.8877 | 0.4973 | 0.5192 | 0.6422 | 0.6562 | 0.6522 | -0.5946 | 0.5637 | 0.5797 |
| **Mpnet** | 0.6378 | 0.8877 | 1.0000 | 0.5246 | 0.4971 | 0.6319 | 0.6503 | 0.6519 | -0.5803 | 0.5518 | 0.5687 |
| **Roberta** | 0.6737 | 0.4973 | 0.5246 | 1.0000 | 0.5228 | 0.6636 | 0.6810 | 0.6923 | -0.6094 | 0.5794 | 0.5960 |
| **Leven** | 0.6334 | 0.5192 | 0.4971 | 0.5228 | 1.0000 | 0.7796 | 0.7635 | 0.7385 | -0.8634 | 0.8220 | 0.8330 |
| **Jaccard** | 0.7915 | 0.6422 | 0.6319 | 0.6636 | 0.7796 | 1.0000 | 0.9876 | 0.9323 | -0.8745 | 0.8710 | 0.8820 |
| **Dice** | 0.8083 | 0.6562 | 0.6503 | 0.6810 | 0.7635 | 0.9876 | 1.0000 | 0.9495 | -0.8705 | 0.8372 | 0.8540 |
| **Cosine** | 0.8221 | 0.6522 | 0.6519 | 0.6923 | 0.7385 | 0.9323 | 0.9495 | 1.0000 | -0.8532 | 0.8226 | 0.8393 |
| **TER** | -0.7357 | -0.5946 | -0.5803 | -0.6094 | -0.8634 | -0.8745 | -0.8705 | -0.8532 | 1.0000 | -0.8305 | -0.8651 |
| **CHRF** | 0.7034 | 0.5637 | 0.5518 | 0.5794 | 0.8220 | 0.8710 | 0.8372 | 0.8226 | -0.8305 | 1.0000 | 0.9858 |
| **BLEU** | 0.7234 | 0.5797 | 0.5687 | 0.5960 | 0.8330 | 0.8820 | 0.8540 | 0.8393 | -0.8651 | 0.9858 | 1.0000 |

We further calculated the arithmetic mean of Pearson correlation coefficients on the 20 datasets of sentence pairs. As shown in Table 3, the average Pearson correlation coefficient is almost the same as the Pearson correlation coefficient in each dataset of sentence pairs, which further indicates that this is an inherent attribute of the Chinese language itself.

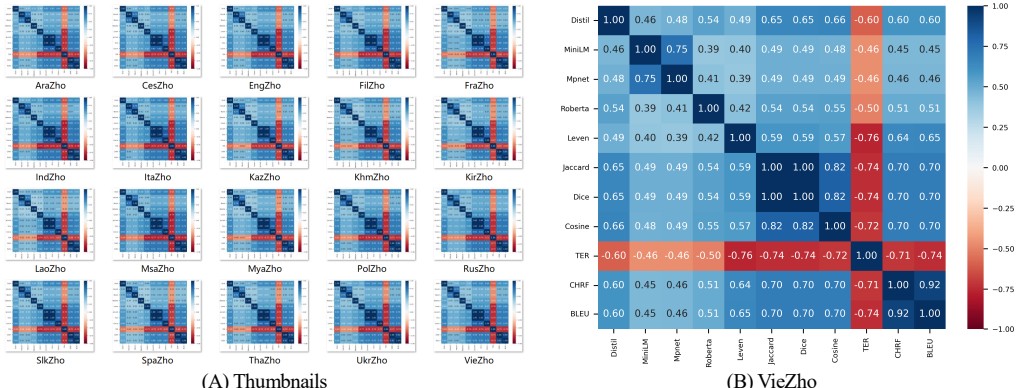

(A) Thumbnails        (B) VieZho

Figure 4: Kendall Correlation Coefficient Heatmap in Chinese

Table 4: Average Kendall Correlation Coefficient in Chinese

| | Distil | MiniLM | Mpnet | Roberta | Leven | Jaccard | Dice | Cosine | TER | CHRF | BLEU |
|---|---|---|---|---|---|---|---|---|---|---|---|
| **Distil** | 1.0000 | 0.4533 | 0.4721 | 0.5238 | 0.4830 | 0.6339 | 0.6339 | 0.6497 | -0.5887 | 0.5859 | 0.5874 |
| **MiniLM** | 0.4533 | 1.0000 | 0.7352 | 0.3745 | 0.3898 | 0.4825 | 0.4825 | 0.4767 | -0.4562 | 0.4441 | 0.4462 |
| **Mpnet** | 0.4721 | 0.7352 | 1.0000 | 0.3940 | 0.3820 | 0.4823 | 0.4823 | 0.4801 | -0.4519 | 0.4461 | 0.4481 |
| **Roberta** | 0.5238 | 0.3745 | 0.3940 | 1.0000 | 0.4062 | 0.5264 | 0.5264 | 0.5336 | -0.4853 | 0.4867 | 0.4877 |
| **Leven** | 0.4830 | 0.3898 | 0.3820 | 0.4062 | 1.0000 | 0.5850 | 0.5850 | 0.5609 | -0.7524 | 0.6315 | 0.6436 |
| **Jaccard** | 0.6339 | 0.4825 | 0.4823 | 0.5264 | 0.5850 | 1.0000 | 1.0000 | 0.8085 | -0.7302 | 0.6990 | 0.6962 |
| **Dice** | 0.6339 | 0.4825 | 0.4823 | 0.5264 | 0.5850 | 1.0000 | 1.0000 | 0.8085 | -0.7302 | 0.6990 | 0.6962 |
| **Cosine** | 0.6497 | 0.4767 | 0.4801 | 0.5336 | 0.5609 | 0.8085 | 0.8085 | 1.0000 | -0.7047 | 0.6948 | 0.6903 |
| **TER** | -0.5887 | -0.4562 | -0.4519 | -0.4853 | -0.7524 | -0.7302 | -0.7302 | -0.7047 | 1.0000 | -0.7037 | -0.7336 |
| **CHRF** | 0.5859 | 0.4441 | 0.4461 | 0.4867 | 0.6315 | 0.6990 | 0.6990 | 0.6948 | -0.7037 | 1.0000 | 0.9185 |
| **BLEU** | 0.5874 | 0.4462 | 0.4481 | 0.4877 | 0.6436 | 0.6962 | 0.6962 | 0.6903 | -0.7336 | 0.9185 | 1.0000 |

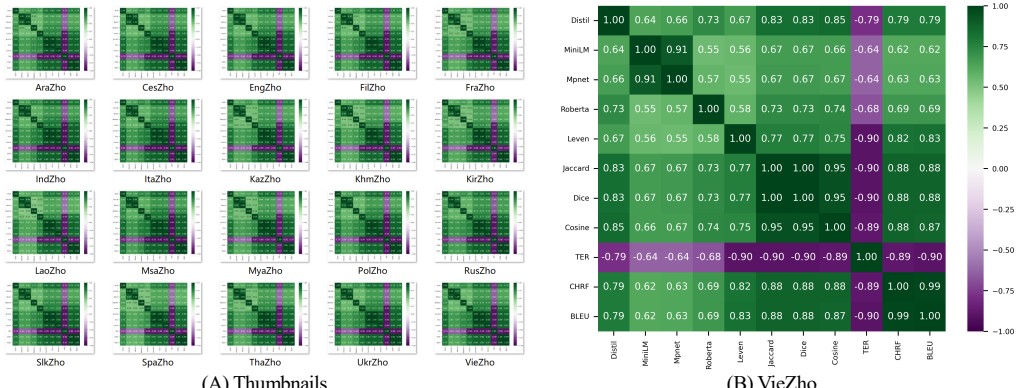

(A) Thumbnails             (B) VieZho

Figure 5: Spearman Correlation Coefficient Heatmap in Chinese

Table 5: Average Spearman Correlation Coefficient in Chinese

|  | Distil | MiniLM | Mpnet | Roberta | Leven | Jaccard | Dice | Cosine | TER | CHRF | BLEU |
|---|---|---|---|---|---|---|---|---|---|---|---|
| **Distil** | 1.0000 | 0.6282 | 0.6496 | 0.7098 | 0.6599 | 0.8192 | 0.8192 | 0.8344 | -0.7761 | 0.7734 | 0.7759 |
| **MiniLM** | 0.6282 | 1.0000 | 0.9006 | 0.5294 | 0.5488 | 0.6623 | 0.6623 | 0.6561 | -0.6310 | 0.6164 | 0.6191 |
| **Mpnet** | 0.6496 | 0.9006 | 1.0000 | 0.5537 | 0.5385 | 0.6615 | 0.6615 | 0.6594 | -0.6253 | 0.6180 | 0.6208 |
| **Roberta** | 0.7098 | 0.5294 | 0.5537 | 1.0000 | 0.5674 | 0.7097 | 0.7097 | 0.7186 | -0.6626 | 0.6641 | 0.6663 |
| **Leven** | 0.6599 | 0.5488 | 0.5385 | 0.5674 | 1.0000 | 0.7607 | 0.7607 | 0.7393 | -0.8945 | 0.8098 | 0.8167 |
| **Jaccard** | 0.8192 | 0.6623 | 0.6615 | 0.7097 | 0.7607 | 1.0000 | 1.0000 | 0.9414 | -0.8973 | 0.8739 | 0.8727 |
| **Dice** | 0.8192 | 0.6623 | 0.6615 | 0.7097 | 0.7607 | 1.0000 | 1.0000 | 0.9414 | -0.8973 | 0.8739 | 0.8727 |
| **Cosine** | 0.8344 | 0.6561 | 0.6594 | 0.7186 | 0.7393 | 0.9414 | 0.9414 | 1.0000 | -0.8790 | 0.8719 | 0.8695 |
| **TER** | -0.7761 | -0.6310 | -0.6253 | -0.6626 | -0.8945 | -0.8973 | -0.8973 | -0.8790 | 1.0000 | -0.8762 | -0.8984 |
| **CHRF** | 0.7734 | 0.6164 | 0.6180 | 0.6641 | 0.8098 | 0.8739 | 0.8739 | 0.8719 | -0.8762 | 1.0000 | 0.9905 |
| **BLEU** | 0.7759 | 0.6191 | 0.6208 | 0.6663 | 0.8167 | 0.8727 | 0.8727 | 0.8695 | -0.8984 | 0.9905 | 1.0000 |

We also performed Kendall correlation analysis and Spearman correlation analysis on the above 11 variables and calculated the Kendall correlation coefficients and Spearman correlation coefficients between any two variables in the {Distil, MiniLM, Mpnet, Roberta, Leven, Jaccard, Dice, Cosine, TER, CHRF, BLEU} set respectively. Figure 4 is the Kendall correlation coefficient heatmap, and Table 4 is the average Kendall correlation coefficient. Figure 5 is the Spearman correlation coefficient heatmap, and Table 5 is the average Spearman correlation coefficient. The experimental results of Kendall correlation analysis and Spearman correlation analysis are consistent with those of Pearson correlation analysis.

## 4.3 EXPERIMENTAL RESULTS IN 20 LANGUAGES

In the same way, we also ran MT from Chinese to language X on the above-mentioned 20 XZho datasets of sentence pairs, and performed correlation analysis experiments on 11 variables in the {Distil, MiniLM, Mpnet, Roberta, Leven, Jaccard, Dice, Cosine, TER, CHRF, BLEU} set, trying to explore the morphological and semantic characteristics of these 20 languages. Finally, the Pearson, Kendall, and Spearman correlation coefficient heatmaps shown in Figure 6 were drawn.

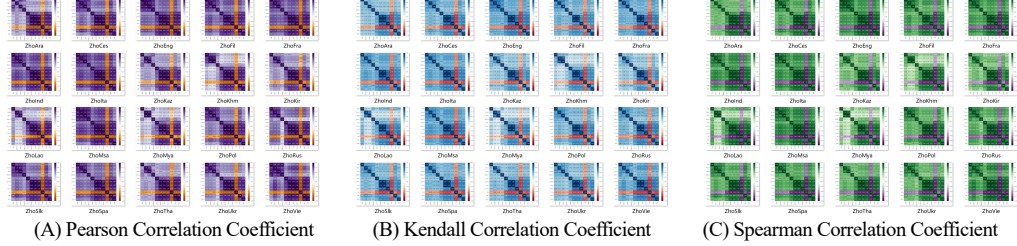

(A) Pearson Correlation Coefficient    (B) Kendall Correlation Coefficient    (C) Spearman Correlation Coefficient

Figure 6: Correlation Coefficient Heatmaps in 20 Languages

Observing the heatmaps of the above 20 languages, it is found that the correlation among the morphological evaluation metrics or that among the semantic evaluation metrics of each language is significant, while the correlation between the morphological and semantic evaluation metrics is quite different. We straightforwardly divided it into three grades according to the values of correlation from high to low. The first grade: Latin alphabet or similar Latin alphabet languages, includes Ces, Eng, Fil, Fra, Ind, Ita, Msa, Pol, Slk, Spa, and Vie, a total of 11 languages; The second grade: Arabic alphabet and Cyrillic alphabet languages, includes Ara, Kaz, Kir, Rus, and Ukr; The last grade: non-universal alphabet language, includes Khm, Lao, Mya, and Tha. Further analysis shows that the value of the correlation coefficient is approximately proportional to the morphological processing ability of the corresponding language in the experimental MT system.

To sum up, in order to overcome the personality deviation of manual evaluation and improve the evaluation efficiency, people put forward a morphological evaluation method based on a manual-translated reference sentence. Because of the diversity of human language, especially the large number of synonyms, people have to provide multiple manual-translated reference sentences to alleviate it. Unfortunately, so far, most MT evaluation datasets only provide one manual-translated reference sentence. With the emergence of pre-trained models, people can basically implement the processing of synonymous sentences with only one manual-translated reference sentence through semantic similarity matching in big data space. We believe that this so-called semantic evaluation is nothing more than more ingenious morphological statistics in the big data space.

## 5   CONCLUSION

This paper focuses on the issue of MT evaluation and quantitatively analyzes the correlation between 7 morphological evaluation metrics and 4 semantic evaluation metrics by using the Pearson correlation coefficient, the Kendall correlation coefficient, and the Spearman correlation coefficient. The analysis results of 21 languages show that for any language, there is a strong correlation among various evaluation metrics of the language, and the so-called deep "semantics" is just another high-level "morphology" under the current Turing computing system. Among different languages, the correlation between morphological evaluation metrics and semantic evaluation metrics is significantly different, and the value of the same correlation coefficient is approximately proportional to the morphological processing ability of the corresponding language. At present, our experimental MT system has a decreasing morphological processing ability of the Latin alphabet or similar Latin alphabet languages, the Arabic alphabet and Cyrillic alphabet languages, and non-universal alphabet languages. We believe that the ability to deal with the inherent morphological attributes of the language determines the translation effect of the MT model on the inherent semantic attributes of the language.

Future research is devoted to expanding the number of languages for quantitative analysis, breaking through the limitation of the MT domain, and studying the quantitative relationship between language morphology and semantics at a more general level. Since the morphology and semantics of a language are closely related, what are those irrelevant parts? We found that the so-called semantics of pre-trained models is just morphology, so can we further guess that "The semantics of language do not exist at all?"

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
