# OpenReview forum: "Correlation Analysis of Evaluation Metrics for Machine Translation"
_ICLR.cc/2025/Conference — Submitted to ICLR 2025_

### Official Review · Reviewer_LGfC · 2024-10-30

**Soundness:** 2
**Presentation:** 3
**Contribution:** 2
**Rating:** 5
**Confidence:** 5

**Summary:**

This paper analyzes the correlation between evaluation metrics for machine translation, specifically focusing on seven commonly used morphological evaluation metrics and four semantic evaluation metrics based on pre-trained models. The study claimed that there is a strong correlation between morphological evaluation metrics, as well as between semantic evaluation metrics. The main contribution of this paper is providing insights into the relationship between different evaluation metrics and suggesting personalized selection methods for evaluation metrics.

**Strengths:**

This paper provides an insightful analysis of the correlation between evaluation metrics for machine translation. The study systematically analyzed the correlation between seven morphological evaluation metrics and four semantic evaluation metrics, and  reveals a strong correlation among different morphological evaluation metrics. This research contributes to the understanding of how different evaluation metrics relate to each other and suggests personalized selection methods based on language characteristics.

In terms of quality, the authors conduct a thorough analysis of the correlation coefficients obtained through Pearson, Kendall, and Spearman tests, providing comprehensive results and supporting their conclusions.

Clarity is another strength of this paper. The authors present their findings in an organized manner and provide clear explanations of their methodology and results. They also use appropriate visual aids like figures and tables to help readers understand their findings better.

**Weaknesses:**

The paper inadequately presents the results. It only shows thumbnails of ZhoX without enlarged images, making specific values difficult to discern.
The paper only presents results for VieZho. It is recommended to include additional language results  in the Appendix.
The paper has a strong start and weak finish. The value of the work will be much improved if the authors conduct a more detailed analysis of why these metrics perform differently across various languages and which types of languages are better suited to specific metrics.
In linguistics, semantics and morphology have their own definitions and functions, and there is naturally a connection between them. The current work is insufficient to support the conclusion that “the semantics of language do not exist.”

**Questions:**

1. In Figure 3, some of the Pearson score of semantic-based metrics are under 0.6. It is not confident to say the correlation is strong. (A Strong correction of Pearson score is usually above 0.6). And since the result is from VieZho, Chinese is as the target language, character-Chinese has less morphological errors, is that a reason the morphology-based metrics have better correlation? That is also one reason I am curious about the detailed score of ZhoX.

---

### Official Review · Reviewer_8pdt · 2024-11-02

**Soundness:** 1
**Presentation:** 1
**Contribution:** 1
**Rating:** 1
**Confidence:** 5

**Summary:**

The paper investigates correlation coefficients between different automatic scores, some of them used for machine translation. A machine translation system was developed for translation from Chinese into 20 different languages, automatic scores were calculated, and three correlation coefficients were investigated: Pearson's, Spearman's and Kendall's.

**Strengths:**

A thorough investigation of different automatic metrics is important.

**Weaknesses:**

The authors should familiarise themselves with machine translation and its evaluation.

The most important shared task involving many machine translation systems, language pairs and evaluation metrics:
https://www2.statmt.org/wmt24/

To name some of the problems:

WMT shared task is not at all mentioned/cited

the division of the metrics in groups is incorrect: there is no "morphological" and "semantic" metrics, only overlap/match based metrics and neural metrics based on embeddings/representations

while some widely used automatic metrics are included, some of the automatic scores are not at all used as machine translation evaluation metric

the analysis was carried out using one single MT system, so the behaviour of the metrics for comparing different systems is not tested at all

the definition of machine translation is incorrect "uses a target natural language form to paraphrase the semantic of a source natural language"

**Questions:**

no particular questions, the authors should familiarise themselves with the WMT shared tasks

---

### Official Review · Reviewer_rMeT · 2024-11-03

**Soundness:** 2
**Presentation:** 2
**Contribution:** 2
**Rating:** 3
**Confidence:** 4

**Summary:**

This work looks at understanding the relation between machine translation metrics that operate at the surface level and metrics that try to operate at the sentence meaning level. The setup consists of 40 language pairs (XXtoZh and ZhtoXX directions) where MT models are trained from scratch and outputs from these models are evaluated with different MT metrics. This is followed by correlating the scores between different metrics per language pair. Analysis of how different metrics operate is carried out based on these correlation scores.

**Strengths:**

1. The choice of languages within language pairs is extensive. Further, by keeping Zh as the central language, new insights about MT evaluation can be uncovered.
2. The use of semi-supervised data augmentation to build MT models is quite interesting. As all of the data and models are in-house, the setup can be useful for carefully analysis

**Weaknesses:**

1. While the setup contains 11 different evaluation strategies, the proposed experiment setup may need further validation to ensure their reliability (please see details in the next section).
2. Contemporary metrics like COMET (https://arxiv.org/abs/2009.09025), UniTE (https://aclanthology.org/2022.acl-long.558/), or even BERTScore (https://arxiv.org/abs/1904.09675), that is closest to the mentioned setup for semantic similarity have not been included in the analysis
3. Previous literature has not been appropriately cited.

**Questions:**

Questions/Comments

1. The sentence embeddings used for semantic similarity measurement are currently the monolingual English versions. Could you please explain why these were used instead of their multilingual equivalents, considering that the language pairs do contain non English languages. Did you observe a lot of unknown tokens in the tokenizer state before calculating the semantic similarity for outputs in languages other than English?

2. The correlation in MT evaluation is generally carried out with the output scores of the metrics and corresponding human judgment to understand how well metrics mimic human evaluation. Could you please explain the setup for correlating metric scores with other metric scores? Generally, it is very likely that the metrics from a particularly family have similar scores because they make similar predictions and similar mistakes (See https://arxiv.org/abs/2401.16313)
Re the correlation being similar for morphological and semantic metrics, I am wondering if the results are so because of the previous point. It would be useful to also report raw scores of the individual metrics, to check if some unavoidable bug has not been introduced in the setup. If the setup is bug-free, it is a useful result.

3. Some qualitative analysis of the outputs can also strengthen the claims.


Other changes.

4. Please use the ISO language codes https://en.wikipedia.org/wiki/List_of_ISO_639_language_codes

5. Please release code, datasets, experiment results for better reproducibility. As the datasets/models used in this work are not based in the previous literature, it would be helpful to look at the data before making the claims in the last section of the conclusion

6. While the conclusion raises interesting points about semantic understanding in deep learning models, it may benefit from additional empirical support to strengthen these claims. Further, the abstract and the conclusion mention "Turing computing systems" without any explanation of the term.

7. Sections 2 and 3 offer detailed explanations of metrics which can be useful but a lot of the metrics are well studied, hence, that sections can be respectively shortened.

8. Abstract: Replace the term "foreign" with "distinct". The use of the term "foreign" can be problematic to some readers.

9. Results must be reported up to 3 significant digits. Averaging of results can lead to spurious claims.

10. Missing references (not mentioned in the previous text) -

Markus Freitag, Nitika Mathur, Chi-kiu Lo, Eleftherios Avramidis, Ricardo Rei, Brian Thompson, Tom Kocmi, Frederic Blain, Daniel Deutsch, Craig Stewart, Chrysoula Zerva, Sheila Castilho, Alon Lavie, and George Foster. 2023. Results of WMT23 Metrics Shared Task: Metrics Might Be Guilty but References Are Not Innocent. In Proceedings of the Eighth Conference on Machine Translation, pages 578–628, Singapore. Association for Computational Linguistics.

Nitika Mathur, Timothy Baldwin, and Trevor Cohn. 2020. Tangled up in BLEU: Reevaluating the Evaluation of Automatic Machine Translation Evaluation Metrics. In Proceedings of the 58th Annual Meeting of the Association for Computational Linguistics, pages 4984–4997, Online. Association for Computational Linguistics.

Tom Kocmi, Eleftherios Avramidis, Rachel Bawden, Ondřej Bojar, Anton Dvorkovich, Christian Federmann, Mark Fishel, Markus Freitag, Thamme Gowda, Roman Grundkiewicz, Barry Haddow, Philipp Koehn, Benjamin Marie, Christof Monz, Makoto Morishita, Kenton Murray, Makoto Nagata, Toshiaki Nakazawa, Martin Popel, et al.. 2023. Findings of the 2023 Conference on Machine Translation (WMT23): LLMs Are Here but Not Quite There Yet. In Proceedings of the Eighth Conference on Machine Translation, pages 1–42, Singapore. Association for Computational Linguistics.

---

### Official Review · Reviewer_iZ2z · 2024-11-04

**Soundness:** 3
**Presentation:** 3
**Contribution:** 1
**Rating:** 1
**Confidence:** 4

**Summary:**

This paper conducts a correlational analysis of the machine translation evaluation metrics. The metrics of interest includes "morphological" metrics which covers metrics that checks for surface-form similarity, such as BLEU and chrF, and "semantic" metrics, which basically computes the similarity of sentence embeddings between the translation and the reference. The study finds very strong correlations with morphological metrics, while also showing that semantic metrics also have strong correlations when different semantic embeddings are used, and also with the morphological metrics. The study on 20 Chinese-centric language pairs also show that correlations between semantic vs. morphological metrics also vary highly between different language pairs.

**Strengths:**

1. This is a thorough study that covers many different correlation measures, metrics variations, as well as language pairs.
2. The data setup is a Chinese-centric one, which offers a novel perspective on top of many existing English-centric studies.

**Weaknesses:**

1. At its current shape, the paper unfortunately doesn't add much to the existing literature, mainly because the paper seems detached in many ways from the state-of-the-art of the machine translation metrics research, elaborated in the following aspects:
1.1. The authors made no mentions at all about the findings of the past WMT metrics shared tasks (most recent: http://www2.statmt.org/wmt24/metrics-task.html), which overlaps heavily with the topic of study here.
1.2. While the choice of morphological metrics seems reasonable, the choice of semantic metrics seem very unorthodox. I would suggest the authors consider some more frequently used options: (1) COMET (https://arxiv.org/abs/2009.09025) (2) BERTScore (https://arxiv.org/abs/1904.09675) (3) BLEURT (https://arxiv.org/abs/2004.04696) (4) MetricX (https://aclanthology.org/2023.wmt-1.63.pdf). (5) AutoMQM (https://arxiv.org/abs/2308.07286). The aforementioned shared task is also a good starting point to find more frequently used metrics.
2. A lot of the key questions on how the evaluation is conducted is unclear, to name a few: specific implementation of metrics such as BLEU/ChrF, details of the MT systems used, source of the Chinese-centric evaluation data (e.g. what domain/website were those crawled from?). I listed some of them in the question section.
3. The presentation could be more focused on the contribution of the study, rather than re-introducing concepts such as correlation coefficients (note that you SHOULD mention important implementation details like how the ties are treated in Kendall's tau correlation).

**Questions:**

- a. How are the ties treated when computing Kendall's tau correlation? Those can influence the numbers by a lot. I don't think this has been specified in the paper.
- b. I'm confused about how BLEU and ChrF was computed on sentence-level. Did the author use some version of sentence-level BLEU & ChrF? Those are different (https://github.com/mjpost/sacrebleu/issues/98).
- c. Any chance the Chinese-centric dataset could be released?

---

### Meta-Review · Area_Chair_SpdF · 2024-12-17

**Metareview:**

The paper provides an in-depth analysis of the correlation between machine translation evaluation metrics, focusing on morphological and semantic aspects. The research involves 40 language pairs, particularly emphasizing Chinese-centric language pairs, to evaluate how these metrics correlate. The study reveals strong correlations within morphological metrics, such as BLEU and chrF, and significant correlations within semantic metrics when different embeddings are utilized.

However, the reviewers find that the paper lacks novelty and needs to incorporate widely used metrics such as COMET and BERTScore in its experiments. Furthermore, the presentation of related work needs to be more comprehensive, and the authors should provide sufficient details regarding the experimental setup.

In addition, the author did not reply.

**Additional Comments On Reviewer Discussion:**

there was no rebuttal

---

### Decision · Program_Chairs · 2025-01-22

Reject